# Prevalence, intensity and associated risk factors of *Schistosoma mansoni* infections among schoolchildren around Lake Tana, northwestern Ethiopia

**Tamirat Hailegebriel**[1,2]*, **Endalkachew Nibret**[1,2], **Abaineh Munshea**[1,2], **Zena Ameha**[3]

**1** Department of Biology, College of Science, Bahir Dar University, Bahir Dar, Ethiopia, **2** Biotechnology Research Institute (BRI), Bahir Dar University, Bahir Dar, Ethiopia, **3** Amhara Public Health Institute (APHI), Bahir Dar, Ethiopia

* tamiratbdu@gmail.com

**Data Availability Statement:** The authors confirm that all data underlying the findings are fully

## Abstract

### Background

Schistosomiasis is one of the widely distributed neglected tropical diseases. It is a serious public health problem in sub-Saharan Africa. The disease is highly prevalent and widely distributed in Ethiopia due to suitable environmental factors and human activities. The prevalence and infection intensity varied from locality to locality in the country. This study aimed to assess the prevalence and intensity of *S. mansoni* infection and associated risk factors among schoolchildren around Lake Tana.

### Methods

A school-based cross-sectional study was conducted among 710 schoolchildren from February to April 2021 in eight selected primary schools around Lake Tana. A questionnaire was used to collect data on socio-demographic information and potential risk factors of *S. mansoni* infection. After collecting socio-demographic information, students were requested to bring about 2grams of stool specimens for parasitological examination. The collected stool samples were processed using a single Kato-Katz and Ritchie's concentration techniques. The data were analyzed using SPSS software version 23 and factors with a p-value < 0.05 were considered as statistically significant.

### Results

The overall prevalence of *S. mansoni* was 34.9% (95% CI: 31.4–38.7) among schoolchildren in the study area. The eggs per gram (EPG) of stool ranged from 24 to 1659 with arithmetic and geometric mean values of 138.1 EPG and 85.1 EPG, respectively. The majority of *S. mansoni* infections (61.4%) were classified as low infection intensity. Among the different determinant factors being male (AOR = 1.74; 95%CI = 1.233–2.457; P-value = 0.002), bathing habits (AOR = 1.494; 95%CI = 1.013–2.199; P-value = 0.043) and students attending at Qunzela primary school (AOR = 10.545; 95%CI = 3.264–34.067; P-value = 0.001),

available without restriction. All relevant data are within the paper.

**Funding:** We would like to thank Biotechnology Research Institute (BRI), Bahir Dar University, for financial support for this study. The funder had no role in the study design, data collection and analysis, decision to publish, or preparation of the manuscript.

**Competing interests:** The authors have declared that no competing interests exist.

**Abbreviations:** BRI, Biotechnology Research Institute; SPSS, Statistical Package for Social Science; NTD, Neglected Tropical Diseases; UNESCO, United Nations Educational, Scientific and Cultural Organization; EPG, eggs per gram; PGRCSVD, Post Graduate and Research Vice-Dean; WHO, World Health Organization; AOR, Adjusted Odd Ratio; COR, Crud Odd Ratio; CI, Confidence Interval; MDA, Mass Drug Administration.

Alabo primary school (AOR = 3.386; 95%CI = 1.084–10.572; P-value = 0.036) were significantly associated with *S. mansoni* infection.

## Conclusion

This study revealed that more than one-third of schoolchildren were infected by *S. mansoni* in the study area. The majority of the infections were classified as low infection intensity. Being male, bathing habits and schools in which students attended were independent explanatory factors for *S. mansoni* infection. Therefore, integrated control strategies are needed to improve the health conditions of schoolchildren in the study area.

## Author summary

We conducted a school-based cross-sectional study to assess the prevalence and intensity of *Schistosoma mansoni* infection and associated risk factors. Stool samples were collected from 681 schoolchildren and processed using Kato-Katz and Ritchie's concentration techniques. Among those students, 238 (34.9%) were infected with *S. mansoni*. Kato-Katz and Ritchie's concentration detect 220 (92.4%) and 165 (69.3%) of the total positive cases, respectively. Most of the *S. mansoni* infections were categorized as low infection intensity based on egg per gram of stool. Among the potential risk factors assessed; being male, bathing habits in open water and schools in which students attended were independent predictors for *S. mansoni* infection.

## Background

Schistosomiasis is one of the neglected tropical diseases (NTD) that infects about 237 million people in tropical and subtropical regions [1]. More than 90% of the cases are concentrated in African countries [2,3]. Frequent contact with infested water during bathing, swimming, fishing, and washing of cloth is associated with the high prevalence of schistosomiasis. In addition, the suitability of the climate conditions for snail intermediate hosts and poor environmental sanitation contributed to the high endemicity of schistosomiasis in the region. Schistosomiasis is the second most common disease next to malaria in terms of socio-economic and health impact in the tropics [4]. Children, in particular, are vulnerable to schistosomiasis that leads to a tremendous negative effect on child development in the region.

Schistosomiasis is endemic in Ethiopia, where more than 37.3 million people are living in endemic areas and about 5 million people are infected [5,6]. It is one of the major causes of outpatient morbidity in the country [7]. Schistosomiasis is caused by two species namely *S. haematobium* and *S. mansoni* in Ethiopia, the latter being the most prevalent and widely distributed species. The burden and prevalence of intestinal schistosomiasis are significantly varied from area to area depending on the suitability of snail intermediate hosts and the level of environmental sanitation. Parasitological studies showed that the prevalence of *S. mansoni* is ranged from 10% to 92% in Ethiopia [8–11]. The prevalence of *S. mansoni* could reach as high as 90% in the northwester part of Ethiopia [8,12]. The Ethiopian government launched nationwide mass drug administration (MDA) program in 2015 to reduce the burden of soil-transmitted helminths and schistosomiasis from schoolchildren. The program brings a significant impact on the reduction of schistosomiasis in the country [11,13,14]. However, the effect of MDA varied from study to study depending on various factors such as nature of study

population, level of environmental sanitation, availability of snail intermediate host and level of awareness of the community.

The present study was conducted in and around Lake Tana, which is one of the United Nations Educational, Scientific and Cultural Organization (UNESCO) registered as a world heritage site. The area is one of the hotspots for schistosomiasis due to the frequent water contact behaviour of the community, availability of snail intermediate hosts and high level of environmental contamination. Despite these suitable conditions for schistosomiasis, only limited studies are available on the islands of Lake Tana and surrounding areas [15,16]. Moreover, these studies used a formal-ether concentration approach, which has poor sensitivity as well as unable to determine the infection intensity. Because of these reasons, the prevalence of *S. mansoni* was underestimated in the surrounding areas that could result in a prolonged health problem among children. In addition, the above studies were conducted in few schools of the same geographical areas. Thus, the present study was conducted in eight different schools located in different geographical locations of Lake Tana and the surrounding mainland. Therefore, this study aimed to assess the prevalence and intensity of *S. mansoni* as well as associated risk factors among schoolchildren in selected primary schools around Lake Tana.

## Materials and methods

### Ethic statement

This study was conducted after obtaining ethical clearance from the Ethical Review Committee of College of Science, Bahir Dar University with Ref. No. PGRCSVD/155/2020. The objective of the study was explained to the school director, students and their parents/guardians. Students were participated in the study only after receiving written consent from their family/guardians for students below 18 years of age. All the results of the laboratory examination were kept confidential and communicated to the family/guardians of each student. All students who were found positive for *S. mansoni* and other intestinal parasitic infections were treated free of charge in collaboration with the nearby health centers.

### Study area

This study was conducted in eight selected primary schools in four purposively selected districts (Bahir Dar City Administration, Bahir Dar Zuria, Chuahit, Lebokemekem, and North Achefer) around Lake Tana. The selected district and schools were located at different corners of Lake Tana (Fig 1). Lake Tana is located in north-western Ethiopia at 12°00' N and 37°14' E and outside the Great Rift Valley areas. It is the largest lake (covers an area of 3020 km$^2$) in Ethiopia and the major source of the Blue Nile River. The lake consists of 37 islands and peninsulas; some of them even serve for human habitation. Communities living on islands, peninsulas and surrounding areas are intensively used the lake water for their livelihoods (source of drinking water, washing cloth, bathing, irrigation, fishing, recreational activities and other domestic purposes). As a result of these close contacts, the population living on islands/peninsulas and surrounding areas are at risk of human schistosomiasis, particularly for *S. mansoni*. In addition, the availability of freshwater and climatic conditions are suitable for the reproduction and multiplication of snail intermediate hosts.

### Study design

A school-based cross-sectional study was conducted from February to April 2021 to assess the prevalence of *S. mansoni* and associated risk factors among schoolchildren. A multistage purposive sampling approach was used in the study. First, districts (Libokemekem, Bahirdar City

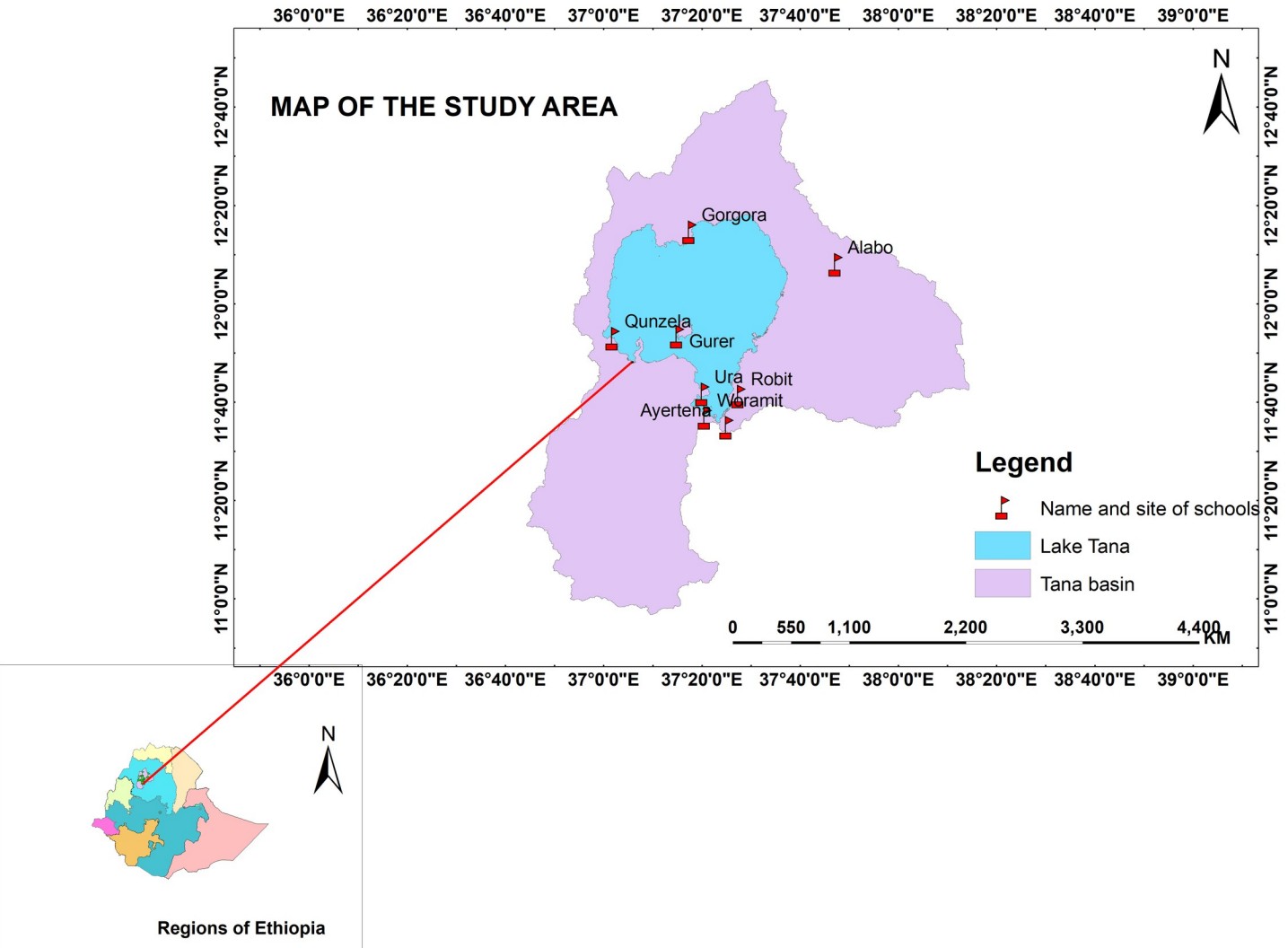

**Fig 1. Map of the study area showing all selected schools around Lake Tana and Tana basin.** The map was prepared using ArcGIS online software.

Administration, Bahir Dar Zuria, Chuahit, and North Achefer) close to Lake Tana were selected purposefully. Second, eight primary schools (Alabo, Ayertena, Gorgora, Gurer, Qunzela, Robit, Woramit and Ura) present in the selected district together with its accessibility were selected. Third, a quota was allocated to each school based on the number of students at each school. Finally, the assigned number to each school was divided into grade levels in each school followed by section/class by considering class roster as a sampling frame. Students were selected by systematic random sampling using intervals obtained by dividing the total source population by the sample size. The first student was obtained by lottery method and the remaining students were selected using the interval by considering the class roster until the last interval of the last section.

## Sources and study population

Children from all schools in the selected islands/peninsula and surrounding areas of Lake Tana were considered as source population while students attending the selected schools in

2020/21 served as the study population. The study was conducted in eight selected primary schools namely Alabo, Ayertena, Gorgora, Gurer, Qunzela, Robit, Woramit and Ura primary schools. The actual number of source population was 7248 (male = 3535 and female = 3713), who attended classes from grade 1 to grade 8 during the 2020/21 academic year.

## Sampling and sample size determination

The sample size was determined using single population proportion formula ($n = Z^2 P (1-P)/d^2$), by considering 30% prevalence of *S. mansoni* [15], 5% marginal error and 95% confidence interval. In addition, the sample size was increased by 5% for the non-response rate. Moreover, taking into consideration the design effect, the sample size was doubled and a total of 710 schoolchildren were invited for socio-demographic information and parasitological examination. The number of students allocated to each school was 129, 127, 72, 31, 90, 141, 95 and 25 from Alabo, Ayertena, Gorgora, Gurer, Qunzela, Robit, Woramit and Ura primary schools, respectively.

## Sample collection and processing

A structured questionnaire was prepared in English and translated to Amahric (local language of the student) to collect socio-demographic information and potential risk factors of *S. mansoni*. Selected school children and/or their parents/guardians were interviewed regarding socio-demographic information and other explanatory variables of schistosomiasis. After completing the interview, the selected children were requested to bring about 2 grams of his/her stool sample using labelled sterile plastic stool caps. The collected stool samples were processed using single Kato-Katz and Ritchie's concentration techniques. Kato-Katz stool smears were employed by taking 41.7mg of stool samples [17]. All eggs of *S. mansoni* were counted from the template and converted to egg per gram of feces (EPG) by multiplying with 24. The infection intensity was classified as light, moderate and heavy based on EPG of 1>99, 100–399 and >400, respectively according to WHO cut-off values [18]. The remaining stool sample was processed using Ritchie's concentration technique [19]. About 0.5 grams of stool sample was placed in a concentration tube that contained 2.5 ml of formalin. The mixture was shaken very well to make it a uniform suspension followed by the addition of 1mL ether. Then, the test tubes were properly mixed and centrifuged at 1500 rpm for three minutes. After discarding the supernatant, the sediments were examined microscopically for the presence of ova and larvae.

## Operational definitions

*Schistosoma mansoni* infection is defined as the observation of *S. mansoni* egg or ova observed in the stool. Infection intensity is used to determine the number of *S. mansoni* egg or ova per gram of feces. Schoolchildren are children who attend primary or secondary schools.

## Quality control

All chemical and consumables were properly checked and quality control was performed at the beginning of the study. The questionnaire was pretested in 5% of students attending nearby schools with similar socio-demographic and environmental conditions but not included in this study. About 10% of the processed samples were selected for re-examination by experienced laboratory technologists who did not have prior knowledge about the result. The result of this examination served as quality control.

## Data analysis

The generated data were processed using SPSS version 23 software to assess the prevalence of *S. mansoni* infection and infection intensity. The infection intensity was categorized as low, moderate and heavy based on EPG according to WHO guidelines. A logistic regression statistical model was used to assess the possible association of factors with *S. mansoni* as well as to determine the strength of the association. Risk factors with a p-value < 0.25 in the univariate analysis were subjected to multivariate logistics regression analysis to control cofounding effects. The magnitude of the association was expressed as odds ratio with a 95% confidence interval and a p-value less than 0.05 was considered statistically significant.

## Results

### Socio-demographic characteristics of study participants

A total of 710 students were invited to participate in the present study. Out of these, 681 schoolchildren (response rate = 95.6%) fulfilled the inclusion criteria and participated in this study. The number of male and female participants were almost equal, 340 (49.9%) and 341 (50.1%), respectively. The age of students ranged from 5 to 22 years (mean age of 11.65±2.66). The largest proportions of students were in the age group of 10–14 years (67%), followers of Orthodox Christianity (97.4%), and lived with their farmer parents/guardians (45.9%). All study participants came from eight primary schools: 140(20.6%), 125(18.4%), 110(16.2%), 90 (13.2%), 88(12.9%), 72(10.6%), 31(4.6%) and 25(3.7%) from Robit, Alabo, Ayertena, Qunzela, Woramit, Gorgora, Gurer and Ura primary schools, respectively (Table 1).

### Prevalence of *Schistosoma mansoni* infection

Out of the 681 study participants, 238(34.9%, 95% CI: 31.4–38.7%) were infected with *S. mansoni*. The prevalence of *S. mansoni* was higher in male students (41.2%) than in female counterparts (28.7%). Similarly, significant differences in *S. mansoni* infections were observed between rural and urban residents and among different family occupations. Moreover, the highest prevalence of *S. mansoni* was observed among students attending Qunzela primary school (68.9%) while the lowest prevalence was detected from students of Ura primary school (16%) (Table 1).

### The Intensity of *Schistosoma mansoni* infections

From the 238 *S. mansoni* infected students, only 220 stool samples were positive by Kato-Katz thick smear, which was used for quantification eggs per template. The minimum and maximum EPG of stool were 24 and 1659, respectively. Likewise, the arithmetic and geometric mean values were 138.1 and 85.1, respectively. According to WHO classification [18], out of 220 students, 135 (61.4%), 75 (34.1%), and 10 (4.5%) had light, moderate and heavy *S. mansoni* infections, respectively. When compared gender-wise, higher moderate *S. mansoni* infection intensity was observed among male students compared to their female counterparts. The intensity of *S. mansoni* infection was higher in students aged 10 to 14 years than in students in other age groups. Similarly, the intensity of *S. mansoni* infection was higher in students attending Qunzela Primary School than in students attending other schools (Table 2).

### Intestinal helminths detected in the study

The present study revealed that eight different helminths were observed among schoolchildren in the study area. *S. mansoni* was the most prevalent followed by hookworm (11.3%), *Ascaris lumbricoides* (9.4%) and *Hymenolepis nana* (2.8%) in the study area. The prevalence of

**Table 1. *Schistosoma mansoni* infections across socio-demographic factors among schoolchildren attending in selected primary schools around Lake Tana, northwest, Ethiopia, 2021.**

| Variables | Infection status of school children | | |
|---|---|---|---|
| | Positive cases (%) | Negative cases (%) | Total (%) |
| **Gender** | | | |
| Male | 140 (41.2) | 200 (58.8) | 340 (49.9) |
| Female | 98 (28.7) | 243 (71.3) | 341 (50.1) |
| **Age (year)** | | | |
| 5–9 | 43 (31.4) | 94 (68.6) | 137 (20.1) |
| 10–14 | 168 (36.8) | 288 (63.2) | 456 (67) |
| 15–19 | 26 (31.7) | 56 (68.3) | 82 (12) |
| >19 | 1 (16.7) | 5 (83.3) | 6 (0.9) |
| **Students grade** | | | |
| One | 28 (33.7) | 55 (66.3) | 83 (12.2) |
| Two | 25 (29.8) | 59 (70.2) | 84 (12.3) |
| Three | 33 (33.7) | 65 (66.3) | 98 (14.4) |
| Four | 32 (39.5) | 49 (60.5) | 81 (11.9) |
| Five | 46 (44.2) | 58 (55.8) | 104 (15.3) |
| Six | 22 (26.5) | 61 (73.5) | 83 (12.2) |
| Seven | 30 (37) | 51 (63) | 81 (11.9) |
| Eight | 22 (32.8) | 45 (67.2) | 67 (9.8) |
| **Grade level** | | | |
| Grade 1–4 | 118 (34.1) | 228 (65.9) | 346 (50.8) |
| Grade 5–8 | 120 (35.8) | 215 (64.2) | 335 (49.2) |
| **Schools name** | | | |
| Alabo | 49 (39.2) | 76 (60.8) | 125 (18.4) |
| Ayertena | 40 (36.4) | 70 (63.6) | 110 (16.2) |
| Gorgora | 21 (29.2) | 51 (70.8) | 72 (10.6) |
| Gurer | 11 (35.5) | 20 (64.5) | 31 (4.6) |
| Qunzela | 62 (68.9) | 28 (31.1) | 90 (13.2) |
| Robit | 36 (25.7) | 104 (74.3) | 140 (20.6) |
| Woramit | 15 (17.0) | 73 (83.0) | 88 (12.9) |
| Ura | 4 (16) | 21 (84) | 25 (3.7) |
| **Resident** | | | |
| Urban | 157 (43.1) | 207 (56.9) | 364 (53.5) |
| Rural | 81 (25.6) | 236 (74.4) | 317 (46.5) |
| **Religion** | | | |
| Orthodox | 232 (35.0) | 431 (65) | 663 (97.4) |
| Muslim | 6 (33.3) | 12 (66.7) | 18 (2.6) |
| **Family occupation** | | | |
| Employed | 35 (37.2) | 59 (62.8) | 94 (13.8) |
| Farmer | 92 (29.4) | 221 (70.6) | 313 (45.9) |
| Merchant | 84 (42.0) | 116 (58.0) | 200 (29.4) |
| Others | 27 (36.5) | 47 (63.5) | 74 (10.9) |
| **Family size** | | | |
| 1–3 | 47 (39.5) | 72 (60.5) | 119 (17.5) |
| 4–6 | 138 (34.0) | 268 (66.0) | 406 (59.6) |
| Above 6 | 53 (34.0) | 103 (66.0) | 156 (22.9) |
| **Total** | **238 (34.9)** | **443 (65.1)** | **681** |

**Table 2. Intensity of *Schistosoma mansoni* infection among schoolchildren using Kato-Katz stool smear technique in selected schools around Lake Tana, northwest, Ethiopia (n = 220), 2021.**

| Variables | Children examined | Light infection No. (%) | Moderate infection No. (%) | Heavy infection No. (%) |
|---|---|---|---|---|
| **Gender** | | | | |
| Male | 133 | 77 (57.9) | 50 (37.6) | 6 (4.5) |
| Female | 87 | 58 (66.7) | 25 (28.7) | 4 (4.6) |
| **Age in years** | | | | |
| 5–9 | 35 | 20 (57.1) | 11 (31.4) | 4 (11.4) |
| 10–14 | 159 | 96 (60.4) | 57 (35.8) | 6 (3.8) |
| 15–19 | 25 | 19 (76.0) | 6 (24.0) | 0 (0) |
| >19 | 1 | 0 (0) | 1 (100) | 0 (0) |
| **Primary schools** | | | | |
| Alabo | 47 | 32 (68.1) | 13 (27.7) | 2 (4.3) |
| Ayertena | 38 | 15 (39.5) | 21 (55.3) | 2 (5.3) |
| Gorgora | 18 | 14 (77.8) | 3 (16.7) | 1 (5.6) |
| Gurer | 8 | 6 (75.0) | 2 (25.0) | 0 (0) |
| Qunzela | 60 | 32 (53.3) | 23 (38.3) | 5 (8.3) |
| Robit | 33 | 22 (66.7) | 11 (33.3) | 0 (0) |
| Woramit | 12 | 10 (83.3) | 2 (16.7) | 0 (0) |
| Ura | 4 | 4 (100) | 0 (0) | 0 (0) |
| **Total** | **220** | **135 (61.4)** | **75 (34.1)** | **10 (4.5)** |

helminths obtained by Kato-Katz, Ritchie's concentration techniques and combined results was presented in Table 3.

## Potential risk factors associated with *S. mansoni* infection

Logistic regression analysis was conducted to identify explanatory variables associated with *S. mansoni* infection. Univariate logistic regression analysis showed that gender of students (p = 0.001), residential area (p<0.001), schools in which students attended (p < 0.001), availability of open water in the nearby area (p = 0.023), swimming habit (p = 0.004), bathing habit (p < 0.001), and washing cloth (p = 0.002) in open water were significantly associated with *S. mansoni* infection (Table 4). However, family occupation, fishing habits, crossing open water, fetching from open water and participating in irrigational activities were not statistically associated with *S. mansoni* infection (p>0.05).

Multivariate logistic regression analysis revealed that students gender, bathing habits and schools in which students attend were explanatory factors for *S. mansoni* infection in the study

**Table 3. Prevalence of *Schistosoma mansoni* and other helminths among schoolchildren obtained using Kao-Katz and Ritchie's concentration techniques around Lake Tana, northwest Ethiopia, 2021.**

| Parasites detected | Ritchie's concentration No. (%) | Kato-Katz smear No. (%) | Combined No. (%) |
|---|---|---|---|
| *Schistosoma mansoni* | 165 (24.2) | 220 (32.3) | 238 (34.9) |
| *Ascaris lumbricoides* | 34 (5) | 47 (6.9) | 64 (9.4) |
| Hookworm | 73 (10.7) | 54 (7.9) | 77 (11.3) |
| *Trichuris trichiura* | 5 (0.7) | 6 (0.9) | 7 (1.0) |
| *Hymenolepis nana* | 15 (2.2) | 10 (1.5) | 19 (2.8) |
| *Strongyloides stercoralis* | 2 (0.3) | - | 2 (0.3) |
| *Enterobius vermicularis* | 4 (0.6) | 2 (0.3) | 5 (0.7) |
| *Taenia* species | 1 (0.1 | - | 1 (0.1) |

**Table 4. Univariate and multivariate analyses of risk factors associated with *S. mansoni* infection among schoolchildren attending primary schools around Lake Tana, northwest, Ethiopia, 2021.**

| Variables | S. mansoni infection status | | | COR (95% CI) | P-value | AOR | P-value |
|---|---|---|---|---|---|---|---|
| | Positive (%) | Negative (%) | Total | | | | |
| **Gender** | | | | | | | |
| Male | 140 (41.2) | 200 (58.8) | 340 | 1.736(1.262–2.388) | 0.001 | 1.741(1.233–2.457) | 0.002* |
| Female | 98 (28.7) | 243 (71.3) | 341 | 1 | | 1 | |
| **Age category** | | | | | | | |
| 5–9 years | 43 (31.4) | 94 (68.6) | 137 | 2.287(0.259–20.176) | 0.456 | | |
| 10–14 years | 168 (36.8) | 288 (63.2) | 456 | 2.917(0.338–25.176) | 0.330 | | |
| 15–19 years | 26 (31.7) | 56 (68.3) | 82 | 2.321(0.258–20.885) | 0.452 | | |
| >19 years | 1 (16.7) | 5 (83.3) | 6 | 1 | | | |
| **Grade** | | | | | | | |
| 1–4 | 118 (34.1) | 228 (65.9) | 346 | 0.927 (0.677–1.271) | 0.639 | | |
| 5–8 | 120 (35.8) | 215 (64.2) | 325 | 1 | | | |
| **Place of residence** | | | | | | | |
| Urban | 157 (43.1) | 207 (56.9) | 364 | 2.210 (1.594–3.064) | 0.000 | 1.444(0.824–2.528) | 0.199 |
| Rural | 81 (25.6) | 236 (74.4) | 317 | 1 | | 1 | |
| **School Name** | | | | | | | |
| Alabo | 49 (39.2) | 76 (60.8) | 125 | 3.385 (1.096–10.457) | 0.034 | 3.386(1.084–10.572) | 0.036* |
| Ayertena | 40 (36.4) | 70 (63.6) | 110 | 3.00 (0.962–9.358) | 0.058 | 3.194(1.009–10.110) | 0.048* |
| Gorgora | 21 (29.2) | 51 (70.8) | 72 | 2.162(0.662–7.062) | 0.202 | 3.932(0.583–6.402) | 0.281 |
| Gurer | 11 (35.5) | 20 (64.5) | 31 | 2.887(0.789–10.573) | 0.109 | 2.475(0.664–9.223) | 0.177 |
| Qunzela | 62 (68.9) | 28 (31.1) | 90 | 11.625(3.649–37.033) | 0.000 | 10.545(3.264–34.067) | 0.001* |
| Robit | 36 (25.7) | 104 (74.3) | 140 | 1.817(0.584–5.651) | 0.302 | 1.723(0.548–5.415) | 0.352 |
| Woramit | 15 (17.0) | 73 (83.0) | 88 | 1.079(0.323–3.600) | 0.902 | 1.027(0.304–3.464) | 0.966 |
| Ura | 4 (16.0) | 21 (84.0) | 25 | 1 | | 1 | |
| **Family size** | | | | | | | |
| 1–3 | 47 (39.5) | 72 (60.5) | 119 | 1 | | | |
| 4–6 | 138 34.0) | 268 (66.0) | 406 | 1.001(0.678–1.478) | 0.997 | | |
| Above 6 | 53 (34.0) | 103 (66.0) | 156 | 1.269(0.773–2.081) | 0.346 | | |
| **Religion** | | | | | | | |
| Orthodox | 232 (35.0) | 431 (65) | 663 | 1.077(0.399–2.906) | 0.884 | | |
| Muslim | 6 (33.3) | 12 (66.7) | 18 | 1 | | | |
| **Mother Education** | | | | | | | |
| Illiterate | 140 (34.3) | 268 (65.7) | 408 | 0.993(0.449–2.192) | 0.985 | | |
| Primary | 58 (37.2) | 98 (62.8) | 156 | 1.124(0.489–2583) | 0.782 | | |
| Secondary | 30 (34.1) | 58 (65.9) | 88 | 0.883(0.406–2.378) | 0.969 | | |
| College | 10 (34.5) | 19 (65.5) | 29 | 1 | | | |
| **Father Education** | | | | | | | |
| Illiterate | 112 (34.6) | 212 (65.4) | 324 | 1.097(0.545–2.210) | 0.795 | | |
| Primary | 64 (33.9) | 125 (66.1) | 189 | 1.063(0.514–2.200) | 0.868 | | |
| Secondary | 49 (38.3) | 79 (61.7) | 128 | 1.288(0.608–2.731) | 0.509 | | |
| College | 13 (32.5) | 27 (67.5) | 40 | 1 | | | |
| **Family occupation** | | | | | | | |
| Farmer | 92(37.2) | 221(62.8) | 313 | 0.702(0.433–1.138) | 0.151 | 1.1017(0.540–1.917) | 0.957 |
| Merchant | 84 (29.4) | 116 (70.6) | 200 | 1.221(0.738–2.020) | 0.438 | 1.164(0.658–2.058) | 0.603 |
| Others | 27 (42.0) | 47 (58.0) | 74 | 0.968(0.515–1.821) | 0.921 | 1.035(0.520–2.060) | 0.923 |
| Employed | 35 (36.5) | 59 (63.5) | 94 | 1 | | | |

(*Continued*)

**Table 4.** (Continued)

| Variables | S. mansoni infection status | | | COR (95% CI) | *P-value* | AOR | *P-value* |
|---|---|---|---|---|---|---|---|
| | Positive (%) | Negative (%) | Total | | | | |
| **Availability of water for swimming** | | | | | | | |
| Yes | 205 (36.9) | 350 (63.1) | 555 | 1.651(1.071–2.545) | 0.023 | 1.016(0.572–1.802) | 0.958 |
| No | 33 (26.2) | 93 (73.8) | 126 | 1 | | 1 | |
| **Swimming habit** | | | | | | | |
| Yes | 157 (39.5) | 242 (60.7) | 399 | 1.610(1.161–2.232) | 0.004 | 1.127(0.674–1.886) | 0.648 |
| | | | | | | 1 | |
| No | 81 (28.7) | 201 (71.3) | 282 | 1 | | | |
| **Bathing habit** | | | | | | | |
| Yes | 173 (39.9) | 261 (60.1) | 434 | 1.856(1.318–2.613) | 0.000 | 1.493(1.013–2.199) | 0.043* |
| No | 65 (26.3) | 182 (73.7) | 247 | 1 | | 1 | |
| **Washing cloth** | | | | | | | |
| Yes | 163 (39.5) | 250 (60.5) | 413 | 1.678(1.204–2.338) | 0.002 | 1.198(0.688–2.087) | 0.523 |
| No | 75 (28.0) | 193 (72.0) | 268 | 1 | | 1 | |
| **Fetching water** | | | | | | | |
| Yes | 55 (34.8) | 103 (65.2) | 158 | 0.992(0.683–1.441) | 0.967 | | |
| No | 183 (35.0) | 340 (65.0) | 523 | 1 | | | |
| **Crossing river** | | | | | | | |
| Yes | 88 (34.5) | 167 (65.5) | 255 | 0.970(0.700–1.343) | 0.853 | | |
| No | 150 (35.2) | 276 (64.8) | 426 | 1 | | | |
| **Fishing habit** | | | | | | | |
| Yes | 26 (41.3) | 37 (58.7) | 63 | 1.346(0.793–2.283) | 0.271 | | |
| No | 212 (34.3) | 406 (65.7) | 618 | 1 | | | |
| **Irrigation involvement** | | | | | | | |
| Yes | 52 (29.5) | 124 (70.5) | 176 | 0.719(0.496–1.042) | 0.082 | 0.885(0.578–1.355) | 0.573 |
| No | 186 (36.8) | 319 (63.2) | 505 | 1 | | 1.418(0.964–2.087) | 0.076 |

**COR**: Crude Odds Ratio; **AOR**: Adjusted Odds Ratio; **CI**: Confidence Interval.

* Significant association

area. Male students were 1.7 fold more likely to be infected with *S. mansoni* than female counterparts (AOR = 1.741, 95% CI: 1.233–2.457, p = 0.002). The odds of infection with *S. mansoni* was 1.5 times higher among students who had a bathing habit in open water than in those without bathing habit (AOR = 1.493; 95% CI: 1.013–2.199, p = 0.043).

School wise, students attending Qunzela primary school were 10.5 times more likely to be infected with *S. mansoni* compared to students from Ura primary school (AOR = 10.545, 95% CI: 3.264–34.067, p = 0.001). Similarly, students attending at Alabo and Ayertena primary schools had 3.4 fold (AOR = 3.386, 95% CI: 1.084–10.572, p = 0.036) and 3.2 fold (AOR = 3.194, 95% CI: 1.009–10.110, p = 0.048) higher likelihood of *S. mansoni* infection compared to students attending in Ura primary school (Table 4).

## Discussion

Epidemiological studies are vital to assess the current *S. mansoni* infection status as well as to evaluate the effectiveness of school-based deworming among schoolchildren. Therefore, this study determined the prevalence and intensity of *S. mansoni* infection and its associated risk factors among schoolchildren. In line with this, the finding of this study serves as a

baseline for designing alternative strategies to improve the health status of schoolchildren in the country.

The present study revealed that 34.9% (95% CI: 31.4–38.7%) of schoolchildren in the study areas were infected with *S. mansoni*. This finding is in line with studies reported from elsewhere [20–26]. In contrast to this finding, higher prevalence of *S. mansoni* were reported from different parts of Ethiopia (58.6% - 89.9%) [8,12,27,28] and those of 62.1%[29], (53.7% and 76.8%) [30,31] and (90.6%) [32] reported from Rwanda, Kenya and Tanzania, respectively. On the other hand, the prevalence of *S. mansoni* observed in the present study was higher than other studies from Ethiopia (15.2% - 27.6%) [33–36], 10.7% from Tanzania [37] and 12.2% from, Kenya [38]. These variations might be associated with differences in the level of environmental sanitation, availability of snail intermediate host, type of diagnostic methods used among studies, level of awareness and water contact behaviour of the target children.

The prevalence obtained in this study is generally classified as moderate prevalence ($\geq$10% but $\leq$50% by parasitological methods) according to WHO guidelines for preventive chemotherapy in human helminthiasis [39]. Although the Ethiopian government launched mass drug administration (MDA) for school-aged children in 2015, the intervention strategy did not reduce the observed prevalence of *S. mansoni* in the study areas. Deworming program is mainly targeting school-aged children and did not target adults, which might serve as sources of infections for other communities. In addition, the deworming program does not prevent from future infection as a result of poor environmental sanitation and high water contact behaviour of schoolchildren in the area. Our previous review showed that 32.5% human population and 15.9% of *Biomphalaria* snails were positive for *S. mansoni* in the country [11]. The findings from this particular review paper suggested the use of integrated strategies such as snail control, proper health education, access to toilet services, and proper environmental sanitation together with the ongoing MDA program.

The present study indicated that the majority of *S. mansoni* infection was classified as light infection, which is in agreement with reports from different parts of Ethiopia: Gorgora town [33], urban setting of south Ethiopia [35], Jimma town [40,41], Mana district of Jimma Zone [42] and from coastal and rainforest zone of Brazil [43] and from selected regions of Gambia [44]. In contrast to the present finding, moderate infection intensity was reported from schoolchildren of Ethiopia: FinchaValley [45], Sanja Area [12], selected areas of central Gondar [46] and from Jinja district of Uganda [47]. The difference in the intensity of *S. mansoni* infection might be associated with the level of environmental sanitation, water contact behaviour of schoolchildren as well as the status and frequency of deworming among the target population.

The prevalence and intensity of *S. mansoni* infection among different age groups were similar in the present study which is in agreement with studies reported from Ejaji [48], Sanja area [12] and Jimma town [41] of Ethiopia. This suggests that all age groups are equally active in their water contact behaviour in the area. In contrast to our finding, students aged 10 to 14 years are at higher risk of schistosomiasis than students below 10 years of age as reported elsewhere [46,49,50]. This might be linked with the high frequency of bathing and swimming habits of schoolchildren of older ages.

The infection intensity of *S. mansoni* was slightly higher in male students than their female counterparts, which is in agreement with other reports from Kemissie and Wondo Genet [9], Jimma town [51] of Ethiopia. The higher infection intensity of *S. mansoni* among males might be associated with various outdoor activities such as fishing, bathing, swimming and irrigation in open water surfaces than female students.

Logistic regression analysis was used to assess the potential associated risk factors considered in the present study and their strength of association with *S. mansoni* infection. The result

of the analysis showed that the gender of students, schools in which they attended and their bathing habits were important explanatory variables of *S. mansoni* infection in the study area.

This study showed a higher prevalence of *S. mansoni* in male students, 41.2% (95% CI: 35.9–46.6) than in female (28.7%) counterparts. The odd of being infected with *S. mansoni* was 1.7 fold higher among male than among female students. Previous studies from different localities have shown that male students are at higher risk of schistosomiasis compared to females [9,34,41,51]. The high prevalence might be associated with a higher frequency of male students contacting with water infested with cercaria for activities such as swimming, bathing, irrigation and fishing activities in open water compared to females. Although females are more involved in fetching water and washing cloth on open water surfaces, opposite results were observed in this study.

A significant variation of *S. mansoni* infection was observed among students attending schools in different geographical areas. Students attending Qunzela primary school were 10.5 times more likely infected with *S. mansoni* compared to those students attending other schools. Similarly, the odds of being infected with *S. mansoni* were 3.4 and 3.2 times higher among students attending Alabo and Ayertena primary schools than among students attending other schools, respectively. This difference might be attributed to the difference in the geographical location of the schools, water contact behaviour, and the status of open space defecation. Qunzela primary school is located on the border of Lake Tana and students have high tendency to swim and bathing in the lake water. Students attending Alabo and Ayertena primary schools are also washing cloth and bathing in the nearby rivers.

Bathing habit is one of the important explanatory risk factors of *S. mansoni* among schoolchildren in the study area. The odds of being infected with *S. mansoni* were 1.5 times higher among students who had a bathing habit in open water than those students without this habit. The association of *S. mansoni* infection and bathing habits of students were documented [12,35,36,44,52]. There will be a high chance of being infected with schistosome cercariae infested water bodies during bathing as far as there is poor environmental sanitation and availability of snail intermediate hosts.

Variables including the availability of open water for swimming, swimming habit and the residential area seemed to be associated with *S. mansoni* infection in univariate analysis. However, all these variables were not served as explanatory variables for *S. mansoni* infection in multivariate analysis.

## Conclusion

This study revealed that more than one-third of schoolchildren were positive for *S. mansoni* infection in the study area. A moderate prevalence of *S. mansoni* infection was observed among schoolchildren around Lake Tana. The majority of infected cases (61.4%) were classified as having low infection intensity. There were significant prevalence variations among different schools. The highest prevalence was observed in children attending Qunzela primary school f while the lowest prevalence was observed in Ura primary school. Being male, bathing habits and attending different schools were important explanatory variables of schistosomiasis infection in the study area. Despite the ongoing school-based deworming program, the prevalence of *S. mansoni* is still high in the studied areas. We recommend an integrated intervention approach that increases community awareness about environmental sanitation and minimizes contact with cercaria infested water. Moreover, snail control measures together with ongoing MDA should be strengthened. Although there was no report about drug resistance, monitoring the effectiveness of praziquantel that is being used for MDA is essential for further action.

The major limitation of this study is the employment of a single Kato-Katz thick stool smear for detection and quantification of *S. mansoni* eggs which might have underestimated the true prevalence and intensity of *S. mansoni* infection among schoolchildren.

## Acknowledgments

We are grateful to students who participated in the study, school directors as well as all other school communities. We also forward our special thanks to Mr Tadesse Hailu for his generous provision of Kato-Katz kits and Ritchie's concentration tubes. Finally, we acknowledge the special supports from Mr Fantahun Taddese, Ms Mastewal Alehegn and Mr Melsew Getaneh during data collection and processing of stool specimens.

### Declarations

We declared that this article is our original work and this manuscript is not submitted to other journals elsewhere.

## Author Contributions

**Conceptualization:** Tamirat Hailegebriel, Endalkachew Nibret, Abaineh Munshea, Zena Ameha.

**Data curation:** Tamirat Hailegebriel, Endalkachew Nibret, Abaineh Munshea, Zena Ameha.

**Formal analysis:** Tamirat Hailegebriel, Endalkachew Nibret, Zena Ameha.

**Funding acquisition:** Tamirat Hailegebriel, Endalkachew Nibret, Abaineh Munshea.

**Investigation:** Tamirat Hailegebriel, Endalkachew Nibret, Abaineh Munshea, Zena Ameha.

**Methodology:** Tamirat Hailegebriel, Endalkachew Nibret, Abaineh Munshea, Zena Ameha.

**Project administration:** Tamirat Hailegebriel, Endalkachew Nibret, Abaineh Munshea.

**Resources:** Endalkachew Nibret, Abaineh Munshea, Zena Ameha.

**Software:** Tamirat Hailegebriel, Endalkachew Nibret.

**Supervision:** Endalkachew Nibret, Abaineh Munshea.

**Validation:** Tamirat Hailegebriel, Endalkachew Nibret, Abaineh Munshea, Zena Ameha.

**Visualization:** Tamirat Hailegebriel, Endalkachew Nibret, Abaineh Munshea, Zena Ameha.

**Writing – original draft:** Tamirat Hailegebriel, Endalkachew Nibret, Abaineh Munshea, Zena Ameha.

**Writing – review & editing:** Tamirat Hailegebriel, Endalkachew Nibret, Abaineh Munshea, Zena Ameha.

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
