## [Decision Letter · Decision Letter 0]

17 Sep 2021

Dear Dr. Hailegebreil,

Thank you very much for submitting your manuscript "Prevalence and intensity of Schistosoma mansoni infections and associated risk factors among schoolchildren around Lake Tana, northwestern Ethiopia" for consideration at PLOS Neglected Tropical Diseases. As with all papers reviewed by the journal, your manuscript was reviewed by members of the editorial board and by several independent reviewers. The reviewers appreciated the attention to an important topic. Based on the reviews, we are likely to accept this manuscript for publication, providing that you modify the manuscript according to the review recommendations. 

Sincerely,

Matty Knight, Ph.D

Associate Editor

Michael Hsieh

Deputy Editor

Reviewer's Responses to Questions

**Key Review Criteria Required for Acceptance?**

**Methods**

-Are the objectives of the study clearly articulated with a clear testable hypothesis stated?

-Is the study design appropriate to address the stated objectives?

-Is the population clearly described and appropriate for the hypothesis being tested?

-Is the sample size sufficient to ensure adequate power to address the hypothesis being tested?

-Were correct statistical analysis used to support conclusions?

-Are there concerns about ethical or regulatory requirements being met?

Reviewer #1: The objectives of the study are clearly articulated. The study design, study population, sample size are appropriate to test the hypothesis and draw conclusion.

Reviewer #2: (No Response)

**Results**

-Does the analysis presented match the analysis plan?

-Are the results clearly and completely presented?

-Are the figures (Tables, Images) of sufficient quality for clarity?

Reviewer #1: Yes. Results are clear and complete with sufficient and quality figures presented.

Reviewer #2: (No Response)

**Conclusions**

-Are the conclusions supported by the data presented?

-Are the limitations of analysis clearly described?

-Do the authors discuss how these data can be helpful to advance our understanding of the topic under study?

-Is public health relevance addressed?

Reviewer #1: Conclusions are supported by the data collected. Part of the conclusion that does not stem from the study is indicated in the attachment. The data public health importance is discussed.

Reviewer #2: (No Response)

**Editorial and Data Presentation Modifications?**

Reviewer #1: The MS needs a minor modification.

Reviewer #2: (No Response)

**Summary and General Comments**

Reviewer #1: The study is important to show the current status of schistosomiasis in the study area. It will give alarm on how to monitor the disease control.

Reviewer #2: (No Response)

PLOS authors have the option to publish the peer review history of their article (what does this mean?). If published, this will include your full peer review and any attached files.

Reviewer #1: No

Reviewer #2: No

Figure Files:

Data Requirements:

Reproducibility:

References

---

## [Editor Report · Decision Letter 1]

28 Sep 2021

Dear Dr. Hailegebreil,

We are pleased to inform you that your manuscript 'Prevalence, intensity and associated risk factors of Schistosoma mansoni infections among schoolchildren around Lake Tana, northwestern Ethiopia' has been provisionally accepted for publication in PLOS Neglected Tropical Diseases.

Best regards,

Matty Knight, Ph.D

Associate Editor

Michael Hsieh

Deputy Editor

---

## [Editor Report · Acceptance letter]

12 Oct 2021

Dear Mr Hailegebreil,

We are delighted to inform you that your manuscript, "Prevalence, intensity and associated risk factors of Schistosoma mansoni infections among schoolchildren around Lake Tana, northwestern Ethiopia," has been formally accepted for publication in PLOS Neglected Tropical Diseases.

Best regards,

Shaden Kamhawi

co-Editor-in-Chief

Paul Brindley

co-Editor-in-Chief
